# Live *Drosophila melanogaster* Larvae Deter Oviposition by *Drosophila suzukii*

**DOI:** 10.3390/insects13080688

**Published:** 2022-07-29

**Authors:** Trisna D. Tungadi, Bethan Shaw, Glen Powell, David R. Hall, Daniel P. Bray, Steven J. Harte, Dudley I. Farman, Herman Wijnen, Michelle T. Fountain

**Affiliations:** 1NIAB, East Malling, West Malling ME19 6BJ, UK; trisnat@gmail.com (T.D.T.); bethan.shaw@niab.com (B.S.); 2RHS Garden Wisley, Woking GU23 6QB, UK; glenpowell@rhs.org.uk; 3Natural Resources Institute, University of Greenwich, Southampton ME4 4TB, UK; d.r.hall@greenwich.ac.uk (D.R.H.); d.bray@greenwich.ac.uk (D.P.B.); s.j.harte@greenwich.ac.uk (S.J.H.); d.i.farman@greenwich.ac.uk (D.I.F.); 4School of Biological Sciences and Institute for Life Sciences, University of Southampton, Southampton SO17 1BJ, UK; h.wijnen@soton.ac.uk

**Keywords:** drosophilidae, integrated pest management, interspecific interactions, push-pull, *sophophora*, repellent, avoidance, behaviour, *drosophila*

## Abstract

**Simple Summary:**

The invasive insect pest, *Drosophila suzukii* Matsumura or spotted-wing drosophila (SWD) lays its eggs in soft and stone fruit. Eggs hatch into larvae, which feed on fruit, causing fruit collapse and significant economic losses worldwide. Current control methods rely primarily on foliar insecticide applications, which are not sustainable long-term solutions. In nature, *D. suzukii* interacts with and encounters other *Drosophila* species, especially towards the end of the growing season when ripening fruits are scarce. We showed previously that *D. suzukii* were deterred from laying eggs on artificial media exposed to egg laying *Drosophila melanogaster*, its sister species. It was hypothesized that a signal was left by *D. melanogaster* which deterred *D. suzukii* from laying eggs. This study aimed to identify from which *D. melanogaster* life stage the egg laying deterrent signal originated and we showed that the presence of live *D. melanogaster* larvae on the egg laying media deter *D. suzukii* from laying eggs. *Drosophila melanogaster* cuticular hydrocarbons were examined as the signal source, but no evidence was found for their involvement. These results have improved our understanding of the interspecific interactions between *D. suzukii* and other *Drosophila* species and could provide new innovative approaches to *D. suzukii* management strategies.

**Abstract:**

The worldwide invasive insect pest, *Drosophila suzukii* Matsumura (spotted-wing *Drosophila*), lays eggs in soft and stone fruit before harvest. Hatched larvae cause fruit collapse and significant economic losses. Current control methods rely primarily on foliar insecticide applications, which are not sustainable long-term solutions due to regulatory restrictions and the risk of insecticide resistance developing. We showed before that *D. suzukii* were deterred from laying eggs on artificial media previously visited by its sister species—*Drosophila melanogaster*. In the current study, laboratory choice test experiments were conducted to identify which *D. melanogaster* life stage (eggs, larvae, or adult) deterred *D. suzukii* oviposition. We demonstrated that the presence of live *D. melanogaster* larvae on the egg-laying media consistently deterred *D. suzukii* oviposition. *Drosophila melanogaster* cuticular hydrocarbons (CHCs) were examined as candidate for the oviposition deterrent. CHCs of larval and adult *D. melanogaster* and *D. suzukii* were analyzed. In both species, the composition of the CHCs of larvae was similar to that of adults, although quantities present were much lower. Furthermore, the CHC profiles of the two species were markedly different. However, when assayed as deterrents in the laboratory choice test experiment, CHC extracts from *D. melanogaster* did not deter oviposition by *D. suzukii*.

## 1. Introduction

*Drosophila suzukii* Matsumura or spotted-wing *Drosophila* (SWD) is a worldwide invasive pest of soft- and stone-fruit. It was first reported from mainland Asia and has spread across the globe over the past two decades [1,2,3,4]. Female *D. suzukii* have an enlarged, serrated ovipositor which can pierce the epicarp of ripening fruit for oviposition (egg laying). Hatched *D. suzukii* larvae feed and spoil fruit from inside, leading to significant economic losses for fruit growers as fruit became damaged and unmarketable [5]. The egg insertion holes are also entry points for secondary pathogens such as fungi and bacteria, which further exacerbate fruit damage [1,6,7]. *Drosophila suzukii* has a wide host range, high fecundity, and capacity to adapt to varying climatic conditions, all of which have contributed to the rapid global spread of this economically important pest [8,9,10]. Currently, the main methods of *D. suzukii* control are insecticide applications, fruit sanitary practices, and barriers of insect-exclusion mesh which are often labour intensive and high in cost [11,12,13,14]. The repeated use of a limited selection of chemical insecticides increases the risk of resistance developing [15,16]. *Drosophila suzukii* resistance to Spinosad has already been recorded in California [17] and Italy [18]. Furthermore, insecticide applications can harm beneficial insects and compromise Integrated Pest Management (IPM) programs [12]. This prompt the need to continually seek alternative and sustainable control methods to optimise pest management for *D. suzukii*. Better understanding of the oviposition behavior of *D. suzukii*, particularly on what attract or deter *D. suzukii* to oviposit and what affects *D. suzukii* oviposition site selection may lead to opportunities to optimize IPM and reduce reliance on chemical insecticides.

In addition to laying eggs in ripening fruit, *D. suzukii* also oviposits and feeds on ripe, damaged, and rotten fruits, especially towards the end of the commercial fruit growing season when most ripening fruits would have been harvested [19,20,21]. This is when *D. suzukii* encounters other *Drosophila* species that oviposit and feed on rotten fruits, such as *Drosophila melanogaster* [5]. The interactions between *D. suzukii* and other Drosophila species in nature are not widely understood. However, in laboratory choice tests, we previously recorded that *D. suzukii* oviposition and offspring emergence was significantly impaired on egg-laying media pre-exposed to *Drosophila melanogaster* flies and containing *D. melanogaster* eggs and larvae, compared to those on unexposed media [22]. Similar findings were published by Kidera and Takahashi [23] whereby *D. suzukii* laid fewer eggs on media previously visited by *D. melanogaster* females. We hypothesized that previous presence of *D. melanogaster* on the egg laying media left a signal or a cue which deter *D. suzukii* oviposition. However, it is not yet known which *D. melanogaster* life stage is the deterrent for *D. suzukii* or where the signal originates from [22,23]. Identifying the source of this deterrent signal from *D. melanogaster* could provide a valuable addition to the limited number of control options available to protect fruit crops from *D. suzukii*. Specifically, if the egg laying deterrent signal can be identified, synthesised, and deployed to optimise current method of control for *D. suzukii.* There are several potential oviposition deterrent signal candidates including insect cuticular hydrocarbons (CHCs) which covers the outer cuticle of insect bodies. Previous work speculated that CHCs play a significant role in the interactions between *D. suzukii* and other *Drosophila* species [22,24,25]. CHCs have multiple functions which includes insect communication and protecting insects from desiccation. The majority of CHCs are non- or semi-volatile long-chain hydrocarbons and act as short-range communication pheromones between insects [26,27,28,29]. *Drosophila suzukii* was able to detect CHCs from other Drosophila species which in turn affect their behavior. For example, perfuming male *D. suzukii* with the sex pheromone of *D. melanogaster*, *cis*-vaccenyl acetate, led to mating disruption [30]. Another study identified four major *D. suzukii* CHCs: 9-tricosene (9-C23:1), 7-tricosene (7-C23:1), 5-tricosene (5-C23:1), and tricosane (*n*-C23) that inhibit mating and courtship in *D. suzukii* [31].

The study presented here aimed to identify which life-stage of *D. melanogaster* was responsible for the oviposition deterrent effect on *D. suzukii*. Different life stages of *D. melanogaster* (mated and virgin adult females and males, eggs and larvae) were tested as candidates for the oviposition deterrent effect to *D. suzukii* in a series of laboratory-based choice test experiments. We also assessed whether CHCs from adult and larvae of both species were the source of the *D. suzukii* oviposition deterrent. Future research directions were discussed.

## 2. Materials and Methods

### 2.1. Drosophila Cultures

Cultures of *D. suzukii* were derived from a strain collected in Trento, Italy, in 2013. They were maintained in glass vials (Kimble Chase 25 × 95 mm Opticlear) containing 5 mL of Bloomington Drosophila Stock Centre (BDSC) standard cornmeal media (in 500 mL distilled H_2_O: 0.5 g Fisher agar, 45 g table sugar, 45 g pre-cooked ground maize, 10 g baker′s yeast, 1 g methylparaben (dissolved in 10 mL of 70% ethanol), 10 g soya flour, 25 g malt extract, and 1 mL propionic acid) and were transferred to new vials every week. Glass Petri dishes filled with BDSC standard cornmeal media were used in all experiments and referred to as “cornmeal media dish”. Cultures of *D. melanogaster* were established from a wild type *D. melanogaster* culture (Blades Biologicals, Kent, UK). *Drosophila suzukii* and *D. melanogaster* cultures were maintained at NIAB, Kent, UK, at 22 °C and 16:8 h light: dark cycle.

### 2.2. General Procedure: Oviposition Choice Test

*Drosophila suzukii* oviposition choice test experiments were conducted using a standardized procedure (Figure 1) as developed by Shaw et al. [22]. Experiments were performed under the same conditions used to maintain cultures at 22 °C and 16:8 h light: dark cycle. All *D. melanogaster* and *D. suzukii* adult flies used for the experiment were 7–10 d old and each experiment consisted of a minimum of 16 replicates unless otherwise mentioned.

Glass Petri dishes (55 mm × 14 mm) were filled with 10 mL of BDSC standard cornmeal media and placed individually into 12 × 7 × 7 cm clear, ventilated Perspex boxes. Blue paper towel sprayed with deionized water lined the base of each box to maintain humidity. *Drosophila melanogaster* flies were anesthetized on a CO_2_ pad (Flystuff, Genesee Scientific, San Diego, CA, USA) for a maximum of 5 min to immobilize them for sexing and counting. After being anesthetized, ten female and five male flies were transferred into each Perspex box containing the cornmeal media dishes. After 15 min, the flies were checked to ensure they had recovered. *Drosophila melanogaster* were left in the box for 48 h for oviposition to occur and then removed from the boxes. The numbers of *D. melanogaster* eggs on each cornmeal media dish were counted under a microscope at ×12 magnification. During 48 h, a proportion of the eggs laid hatched into larvae. Hatched egg cases were visible under the microscope, and these were counted as eggs. Live larvae originated from eggs that hatched during the experiment were not counted and were left on the dish.

In the second stage of the experiment, two cornmeal media dishes were placed in a new Perspex box, one dish containing *D. melanogaster* eggs as described above, and a blank cornmeal media dish which had not been exposed to *D. melanogaster*. Adult *D. suzukii* were anesthetized on the CO_2_ pad and ten females and five males were transferred into each box. After 48 h, *D. suzukii* adults were removed from the box. The numbers of eggs in both cornmeal media dishes were counted under a microscope. On the dish that was unexposed to *D. melanogaster*, all eggs were presumed to be *D. suzukii*. On the dish that was previously exposed to *D. melanogaster*, the number of *D. melanogaster* eggs was subtracted from the total egg count after *D. suzukii* were removed, to give the number of *D. suzukii* eggs laid.

### 2.3. Experiments to Determine Life Stage of Drosophila melanogaster Responsible for Reduction in Oviposition by D. suzukii

#### 2.3.1. Effect of Adult *Drosophila melanogaster* on Oviposition by *D. suzukii*

Adult *D. melanogaster* were removed from culture vials ensuring only juvenile *D. melanogaster* remained. Hourly from 10:00–16:00 newly emerged flies were removed and placed into single sex culture vials containing cornmeal media. Virgin *D. melanogaster* adults were collected over 2 d to gather sufficient numbers to conduct the experiment. These newly emerged male and female flies were left to mature in separate vials for a further 3 d, resulting in flies being 4–5 d old at the start of the experiment. For experiments using virgin male or female *D. melanogaster* as the test stimuli, 15 virgin males or 15 virgin females were given 48 h access to a cornmeal media dish inside a ventilated Perspex box based on the general methodology described above (*n* = 17 and 10 for experiments with virgin females and males, respectively in Figure 2 and Figure 3). In control experiments run simultaneously for this set of experiments, *D. suzukii* were exposed for 48 h to two blank cornmeal media dishes never exposed to flies before. The purpose of this control experiment with two blank dishes was to ensure that *D. suzukii* egg laying decision were not affected by other external factors. In the absence of the stimuli from *D. melanogaster*, we would expect *D. suzukii* to lay similar number of eggs on both blank cornmeal media dishes.

For experiments using mated male or mated female *D. melanogaster* as the stimuli, the same steps were taken to collect the virgin flies. After collection, 6 virgin males and 6 virgin female *D. melanogaster* were placed into the same vial (total *n* = 24 vials) and allowed to mate. After 24 h the flies were anesthetized and separated by sex into separate vials and used for the experiment on the same day. For the experiment, 15 mated female or 15 mated male *D. melanogaster* were allowed 48 h access to a cornmeal media dish and then removed, following the general procedure described above (*n* = 16 for each treatment). For this experiment with mated flies and subsequent experiments (Figure 4, Figure 5, Figure 6, Figure 7 and Figure 8), a control experiment was run simultaneously by offering *D. suzukii* a choice between a dish pre-exposed to *D. melanogaster* or a blank dish not exposed to *D. melanogaster* previously. For the control experiment, we expected *D. suzukii* to lay fewer eggs on dishes pre-exposed to *D. melanogaster* as recorded by Shaw et al. [22].

#### 2.3.2. Effect of *Drosophila melanogaster* Eggs on Oviposition by *D. suzukii*

*Drosophila melanogaster* eggs were collected by exposing cornmeal media Petri dishes to female and male adult *D. melanogaster* for 48 h, after which 30 *D. melanogaster* eggs were transferred individually using a fine paintbrush onto a new cornmeal media dish for the experiment. During transfer, as little media as possible was transferred. A blank dish was added into the Perspex box alongside the dish containing the transferred *D. melanogaster* eggs. Ten female and five male *D. suzukii* were introduced into the box containing dishes with *D. melanogaster* egg and a blank dish. Oviposition by *D. suzukii* was assessed by the general procedure described above (*n* = 20).

In a second experiment, *D. melanogaster* were allowed to lay eggs on the cornmeal media dish for 48 h before the adults were removed. All *D. melanogaster* eggs and larvae were removed using a fine paintbrush before offering the dish to *D. suzukii* alongside a blank dish for the oviposition choice test. Oviposition by *D. suzukii* was assessed according to the general procedure above (*n* = 20).

#### 2.3.3. Effect of *Drosophila melanogaster* Larvae on Oviposition by *D. suzukii*

Live larvae of *D. melanogaster* larvae were collected by exposing cornmeal media Petri dishes to female and male adult *D. melanogaster* to allow for egg-laying and subsequent larval development. After six days, *D. melanogaster* adults were removed and 15 live *D. melanogaster* larvae (equal mix of first, second, and third instar larvae) were transferred individually using a fine paintbrush onto a new cornmeal media dish and left for 1 h to settle. The dish was then paired with a blank cornmeal media dish and oviposition preferences by *D. suzukii* assessed according to the general procedure above (*n* = 18).

In a second experiment, the same methodology was followed, but the *D. melanogaster* larvae were transferred onto a 7 cm diameter Whatman filter paper and killed by placing in a freezer at −20 °C for 4 h. The dead *D. melanogaster* larvae were placed onto cornmeal media dishes (15 per dish) and oviposition preferences by *D. suzukii* assessed according to the general procedure above (*n* = 18). Simultaneous experiments were conducted with live and dead *D. suzukii* larvae following the same method.

#### 2.3.4. Effect of *Drosophila melanogaster* Larval Feeding Activity on Oviposition by *D. suzukii*

*Drosophila melanogaster* larvae were collected as described in the previous section. Fifteen larvae (a mixture of first, second, and third instar) were transferred individually onto a cornmeal media Petri dish using a fine paintbrush and allowed to feed for 48 h. Larvae were then removed individually using fine metal forceps or killed by placing the dish in a freezer at −20 °C for 6 h. These cornmeal media dishes were offered to *D. suzukii* alongside a blank dish and oviposition preferences by *D. suzukii* assessed as above. The same procedures were used with larvae of *D. suzukii,* and the effects on oviposition by *D. suzukii* were also assessed (*n* = 18 for each treatment).

### 2.4. Extraction and Analyses of Cuticular Hydrocarbons (CHCs) from D. melanogaster and D. suzukii

Flies were anesthetized with CO_2_ as described above to be counted and sexed. Ten female or 10 male adult *D. melanogaster* or *D. suzukii* flies were transferred to 1.5 mL glass vials containing 500 µL hexane (Distol Pesticide residue grade, Fisher Scientific) for 10 min at room temperature. Hexane washes were transferred into a new glass vial and stored at −20 °C until analysis. Similar extracts of larvae of *D. melanogaster* and *D. suzukii* were collected using a mixture of first, second, and third instar larvae (*n* = 100) extracted in 500 µL hexane as above.

Collections were analyzed by gas chromatography coupled to mass spectrometry (GC-MS) using a Varian 3700 GC linked directly to a Saturn 2200 ion-trap MS (Varian, now Agilent Technologies, Manchester, UK). Columns (30 m × 0.25 mm i.d. 0.25 μm film thickness) were coated with polar DBWax (Supelco, Gillingham, Dorset, UK) or non-polar VF5 (Varian/Agilent). Injection was splitless (250 °C), the carrier gas was helium (1 mL/min) and the oven temperature was held at 40 °C for 2 min then programmed at 10 °C/min to 250 °C and held for 5 min. Retention Indices (RI) for compounds were calculated relative to the retention times of *n*-alkanes.

Compounds were initially identified from their mass spectra and RI′s on polar and non-polar columns and identifications were confirmed by comparison with synthetic standards. (*Z*)-11-Octadecenyl acetate (cis-vaccenyl acetate, cVA) was available from previous work [32]. (Z)-5-, (*Z*)-7- and (*Z*)-9-tricosenes and pentacosenes were synthesized by Wittig reaction of the appropriate triphenylphosphonium salt and aldehyde in tetrahydrofuran with sodium hexamethylsilazide as base. (*Z,Z*)-7,11-Tricosadiene, pentacosadiene, heptacosadiene and nonacosadiene were synthesized in good yield by a route similar to that used by Billeter et al. (2009) [33]. (*Z,Z*)-6,9-Tricosadiene and pentacosadiene were synthesized by coupling the tosylate of (*Z,Z*)-9,12-octadecadienol, derived from methyl (*Z,Z*)-9,12-octadecadienoate (methyl linoleate), with the appropriate Grignard reagent in tetrahydrofuran in the presence of catalytic dilithium tetrachlorocuprate. 2-Methyltetracosane, and 2-methylhexacosane were synthesized by coupling 2-methylbutyl-magnesium bromide with the tosylates of 1-eicosanol or 1-docosanol, respectively, in tetrahydrofuran in the presence of catalytic dilithium tetrachlorocuprate.

### 2.5. Effect of Cuticular Hydrocarbons (CHCs) from Adults or Larvae of D. melanogaster on Oviposition by D. suzukii

Cuticular hydrocarbon (CHC) extracts were obtained from adult *D. melanogaster* or *D. suzukii* by immersing 80 males or 80 females in 800 µL hexane for 10 min and then transferring the hexane to a clean vial. CHC extracts from larvae were obtained by washing 500 larvae of *D. melanogaster* or *D. suzukii* in 500 µL hexane for 10 min before transferring the extract to a clean vial. A 100 µL aliquot of either wash was pipetted directly onto a cornmeal media dish, which was swirled to ensure that all the surface was covered and then left to air dry for one hour to allow the hexane to evaporate completely. Aliquots of washes were analyzed by GC-MS to confirm that the same compounds were detected as previously from the adult washes. Oviposition choice test experiments comparing dishes treated with CHCs and with 100 µL hexane were performed with *D. suzukii* adults according to the general procedure described above (*n* = 12).

### 2.6. Statistical Analyses

All statistical analyses were performed using R software (R version 3.5.1, R Studio version 1.1.456.). In each experiment, numbers of eggs laid on test and control plates were compared using a Wilcoxon-signed rank test. Figures show differences in number of eggs laid on untreated control plates compared to the number laid on the treated plates. The egg counts raw data for Figure 2, Figure 3, Figure 4, Figure 5, Figure 6, Figure 7, Figure 8, Figure 9, Figure 10 and Figure 11 can be found in Appendix A.

## 3. Results

### 3.1. Experiments to Determine Life Stage of Drosophila melanogaster Responsible for Reduction in Oviposition by D. suzukii

#### 3.1.1. Effect of Adult *Drosophila melanogaster* on Oviposition by *D. suzukii*

There were no significant differences in the numbers of *D. suzukii* eggs recorded on unexposed, blank cornmeal media dishes compared to those pre-exposed to *D. melanogaster* virgin females (Figure 2: Wilcoxon Signed-Rank test *n* = 17, V = 71.5, *p* = 0.83) or virgin males (Figure 3: *n* = 10, V = 31.5, *p* = 0.72). In control experiments run simultaneously to ensure that *D. suzukii* egg laying decision were not affected by external stimuli, two blank dishes that had not been exposed to flies were offered to *D. suzukii*. There were no significant differences in the numbers of eggs laid on these paired blank dishes in either experiment (Figure 2: for control *n* = 17, V = 0.67, *p* = 0.67; Figure 3: for control *n* = 10, V = 19.5, *p* = 0.77).

On cornmeal media pre-exposed to mated *D. melanogaster* males or females only, there were no significant differences in the numbers of eggs laid by *D. suzukii* on dishes that had been exposed to mated females (Figure 4: *n* = 16, V = 54.5, *p* = 0.50) or mated males (Figure 4: *n* = 16, V = 68, *p* = 1.00) in comparison to eggs laid on the unexposed, blank dishes. In a control experiment run simultaneously, *D. suzukii* were presented with a dish that had been pre-exposed to *D. melanogaster* and an unexposed dish. *Drosophila suzukii* laid fewer eggs on the pre-exposed dishes than on dishes that were not exposed to *D. melanogaster* (Figure 4: *n* = 16, V = 120, *p* < 0.01 **) which was expected based on our previous work [22].

#### 3.1.2. Effect of *Drosophila melanogaster* Eggs on Oviposition by *D. suzukii*

There was no significant difference in the number of *D. suzukii* eggs laid on cornmeal media dishes containing transferred *D. melanogaster* eggs compared to the number laid on blank dishes (Figure 5: Wilcoxon Signed-Rank Test, *n* = 20, V = 49.5, *p* = 0.07), suggesting that the presence of *D. melanogaster* eggs alone do not exert an oviposition deterrent effect on *D. suzukii*. In the control treatment run simultaneously, fewer *D. suzukii* eggs were recorded on dishes pre-exposed to *D. melanogaster* compared to those on blank dishes, but on this occasion, the difference was not statistically significant (Figure 5: *n* = 20, V = 154.5, *p* = 0.07).

When *D. melanogaster* was allowed to oviposit for 48 h and then the eggs and larvae laid by *D. melanogaster* were removed before the dishes were offered to *D. suzukii*, there was no significant difference in the numbers of *D. suzukii* eggs laid on dishes where *D. melanogaster* eggs were previously present compared to the number laid on blank dishes (Figure 6: *n* = 20, V = 83, *p* = 0.64). In the control experiment run simultaneously, there were fewer *D. suzukii* eggs on dishes pre-exposed to *D. melanogaster* compared to those on unexposed dishes (Figure 6: *n* = 20, V = 170.5, *p* < 0.01 **).

#### 3.1.3. Effect of *Drosophila melanogaster* Larvae on Oviposition by *D. suzukii*

Using *D. melanogaster* larvae transferred onto the cornmeal media dishes with no adult contact, oviposition by *D. suzukii* was not affected by the presence of dead *D. melanogaster* larvae (Figure 9: Wilcoxon Signed-Rank Test, *n* = 18, V = 60, *p* = 0.28), but fewer *D. suzukii* eggs were recorded on cornmeal media dishes containing live *D. melanogaster* larvae compared to blank dishes (Figure 9: *n* = 18, V = 171, *p* < 0.001 ***). The presence of dead (Figure 9: *n* = 18, V = 51.5, *p* = 0.14) or live *D. suzukii* larvae (Figure 9: *n* = 18, V = 121, *p* = 0.13) did not deter *D. suzukii* from laying eggs.

#### 3.1.4. Effect of *D. melanogaster* Larval Feeding Activity on Oviposition by *D. suzukii*

Transfer of *D. melanogaster* larvae onto a cornmeal media dish where they were left to feed for 48 h before being removed by hand resulted in fewer *D. suzukii* eggs laid on these dishes compared to numbers laid on blank dishes (Figure 10: Wilcoxon Signed-Rank Test, *n* = 18, V = 139.5, *p* < 0.05 *). When cornmeal media dishes were inoculated with larvae of conspecifics, *D. suzukii* laid fewer eggs where *D. suzukii* larvae had been feeding and removed, but this difference was not significant (Figure 10: *n* = 18, V = 129.5, *p* = 0.06. As above, the presence of live larvae of *D. melanogaster* reduced oviposition by *D. suzukii* (Figure 10: *n* = 18, V = 167, *p* < 0.001 ***), but presence of live conspecific larvae did not (Figure 10: *n* = 18, V = 71.5, *p* = 0.45).

When *D. melanogaster* larvae were transferred individually onto a cornmeal media dish and allowed to feed for 48 h before being killed in situ by freezing the dish with the larvae, there was no reduction of *D. suzukii* eggs laid on this dish compared to eggs laid on blank dishes (Figure 11: Wilcoxon Signed-Rank Test, *n* = 18, V = 73.5, *p* = 0.62). Similarly, when dishes were inoculated with larvae of conspecifics killed in situ by freezing, *D. suzukii* did not lay fewer eggs on those dishes compared to blank dishes (Figure 11: *n* = 18, V = 105, *p* = 0.41). However, fewer *D. suzukii* eggs were counted on dishes when live *D. melanogaster* larvae or *D. suzukii* larvae were present (Figure 11: *n* = 18, V = 162.5, *p* < 0.001 *** for *D. melanogaster* live larvae and *n* = 18, V = 32.5, *p* < 0.05 * for *D. suzukii* live larvae).

### 3.2. Effect of Cuticular Hydrocarbons (CHCs) on Oviposition by Drosophila suzukii

Results of analyses of CHCs extracted in hexane from female and male *D. melanogaster* and *D. suzukii* are shown in full in Appendix A and for the main components in Figure 12 below. The most abundant component in all extracts was (*Z*)-7-tricosene (Z7-23H), and there were numerous components in common including saturated hydrocarbons such as heneicosane (21H) and tricosane (23H), other monounsaturated hydrocarbons such as (Z)-9-tricosene (Z9-23H), and 2-methyl substituted hydrocarbons such as 2-methyltetracosane (2Me-24H), 2-methylhexacosane (2Me-26H) and 2-methyloctacosane (2Me-28H) (Figure 12). Compounds only present in extracts from *D. melanogaster* included *cis*-vaccenyl acetate (cVA; (*Z*)-11-octadecenyl acetate, Z11-18Ac) and the (*Z,Z*)-7,11-pentacosadiene, heptacosadiene and nonacosadiene (ZZ7,11-25H, ZZ7,11-27H and ZZ7,11-29H, respectively) [33] (Figure 12). (*Z,Z*)-6,9-Tricosadiene (ZZ6,9-23H) was only present in the extracts from *D. suzukii;* the corresponding pentacosadiene (ZZ6,9-25H) and heptacosadiene (ZZ6,9-27H) were present in extracts from both species in trace amounts. Extracts of *D. suzukii* flies also contained (*Z*)-9-, (*Z*)-7- and (*Z*)-5-nonoacosenes which were not detected in extracts from *D. melanogaster* (Figure 12). (*Z*)-4-Undecenal, a potential sex pheromone of female *D. melanogaster* [34], could not be detected. Amounts of the major component, Z7-23H, in extracts from female and male *D. melanogaster* and *D. suzukii* were 0.5 µg, 0.8 µg, 1.5 µg and 0.6 µg per insect, respectively.

Similar hexane extracts were made of eggs and larvae of both *D. melanogaster* and *D. suzukii.* Analysis by GC-MS showed only traces of volatile material, although Z7-23H could be detected and also ZZ7,11-27H in the case of eggs and larvae from *D. melanogaster.* Amounts of Z7-23H in extracts from eggs and larvae of *D. melanogaster* and *D. suzukii,* were approximately 1 ng per egg or larva. However, it is uncertain whether these compounds were derived from the eggs or larvae or from traces of media which accompanied them during collection. Hexane washes of the cornmeal media dishes after exposure to adult *D. melanogaster* or *D. suzukii* for 48 h, as in the general procedure, showed significant quantities of the CHCs with similar composition to the mixtures obtained directly by extracting adult flies above, with mean amounts of Z7-23H of 1.1 µg and 2.2 µg per plate for adult *D. melanogaster* or *D. suzukii*, respectively (*n* = 3 each).

*Drosophila suzukii* oviposition was not affected on the cornmeal media dishes treated with hexane body washes from *D. melanogaster* or *D. suzukii* adult or larvae compared to dishes treated with hexane only (Figure 7: washes from *D. melanogaster* adult flies Wilcoxon Signed-Rank Test, *n* = 12, V = 46, *p* = 0.61; washes from *D. suzukii* adult flies *n* = 12, V = 30, *p* = 0.50. Figure 8: washes from *D. melanogaster* larvae: *n* = 18, V 73.5, *p* = 0.62; *D. suzukii* larvae *n* = 18, V = 0.71, *p* = 0.54). In the control experiments where dishes were pre-exposed to *D. melanogaster* flies for 48 h before being offered to *D. suzukii*, *D. suzukii* laid fewer eggs on dishes pre-exposed to *D. melanogaster* (Figure 7: *n* = 12, V = 78, *p* < 0.01 **, Figure 8: *n* = 18, V = 169, *p* < 0.001 ***).

## 4. Discussion

Previously we found that *D. suzukii* oviposition and offspring emergence were significantly lower on media pre-exposed to *D. melanogaster* than on unexposed media [22]. We hypothesized that *D. melanogaster* left a signal which deterred *D. suzukii* from ovipositing. This study aimed to determine from which life-stage of *D. melanogaster* this oviposition deterrent signal originates. We tested adult flies, eggs and larvae, but not pupae as these would not have been present during the 96 h of the general experimental procedure. We also tested whether *D. melanogaster* cuticular hydrocarbons (CHCs) were the oviposition deterrent signal for *D. suzukii*.

First, we examined whether the mating status of *D. melanogaster* influences the observed oviposition deterrent effect on *D. suzukii* because *D. melanogaster* used in our previous study had always been mated flies [22]. Mated *D. melanogaster* produces mating pheromones which affect their behavior and alter their CHC profiles [28,29,33,35,36]. Within this study, we demonstrated that previous exposure of the egg laying media to adult *D. melanogaster* males or females, whether mated or unmated, did not deter *D. suzukii* from ovipositing. This suggested that pre-exposure of media to adult *D. melanogaster* flies alone was not enough to deter *D. suzukii* oviposition and the mating status of *D. melanogaster*, per se, was not a determinant in influencing *D. suzukii* egg laying.

We then investigated whether *D. melanogaster* eggs (in the absence of adult cues) were the deterrent signal influencing *D. suzukii* oviposition. We demonstrated that the presence of transferred *D. melanogaster* eggs or the previous presence of eggs on the media did not affect *D. suzukii* oviposition. In our experimental set up where we removed naturally laid *D. melanogaster* eggs before offering the media to *D. suzukii*, if the media contained compounds secreted by *D. melanogaster* females around their egg laying sites, then these appear to not deter *D. suzukii* from laying eggs.

After investigating the effect of eggs, we then examined if *D. melanogaster* larvae were the source of the oviposition deterrent signal. *Drosophila suzukii* laid significantly fewer eggs in the presence of live, but not dead *D. melanogaster* larvae (transferred or killed in situ). In the presence of live larvae of their conspecifics, *D. suzukii* oviposition was not impeded. We placed fifteen *D. melanogaster* larvae of an equal mix of 1st, 2nd, and 3rd instar larvae onto the media for 48 h before offering the media to *D. suzukii*. *Drosophila suzukii* consistently laid fewer eggs on the media containing *D. melanogaster* larvae. However, in one of our experiments (Figure 4), *D. suzukii* was not deterred from laying eggs on media pre-exposed to mated female *D. melanogaster*. In that experimental set up, eggs were laid by mated female *D. melanogaster* during the 48 h, and a proportion of those eggs hatched into 1st instar larvae when the media was offered to *D. suzukii*. Despite this, *D. suzukii* did not lay fewer eggs on this media. It is possible that more eggs are laid by female *D. melanogaster* in the presence of males and/or these and the subsequent larvae are fitter in some way. These factors could lead to larvae causing more profound changes to both chemical and physical composition of the egg laying media, including a decrease in the media firmness. *Drosophila suzukii* tends to prefer firmer and intact substrate to oviposit on which supports this theory [7,37]. *Drosophila* larval density and their foraging activities are also known to alter the surrounding bacterial community and subsequently affected the phenotypical trait of the emerging flies [38]. These considerations suggest there may be multiple signals at play here, possibly context-dependent, to deter oviposition by *D. suzukii* [37].

The cuticular hydrocarbons (CHCs) of *Drosophila* species are known to be species-specific in composition and involved in both intra- and interspecific communication [28]. They were thus prime candidates as potential signal left by *D. melanogaster* that deters oviposition by *D. suzukii*. Analyses of hexane extracts of adult flies confirmed results reported previously for *D. melanogaster* [35] and *D. suzukii* [31], with identities of all the key compounds confirmed with synthetic standards rather than relying only on mass spectra and GC retention indices. Some of the most abundant components were present in CHCs of both species, such as (*Z*)-7- and (*Z*)-9-tricosenes and 2-methyl-hexacosane and octacosane. However, (*Z*)-11-octadecenyl acetate (*cis-*vaccenyl acetate, *c*VA), (*Z,Z*)-7,11-heptacosadiene and (*Z,Z*)-7,11,-nonacosadiene were only present in washes from *D. melanogaster*, and (Z,Z)-6,9-tricosadiene and pentacosadiene were only present in washes from *D. suzukii. c*VA is produced by male *D. melanogaster* and the (Z,Z)-7,11-dienes by females. Our analyses were done on extracts from flies taken from mixed cultures of *D. melanogaster*, reflecting the situation in the bioassay, and these compounds were found in extracts from both sexes, although *c*VA was more abundant in extracts from males and the (Z,Z)-7,11-dienes were more abundant in extracts of females than males.

Moreover, significant amounts of these CHCs were left on the media plates after exposure to the adult flies for 48 h, in line with the results of Farine et al. [29]. Analyses of hexane washes of eggs and larvae of the two species showed traces of the CHCs obtained from the adult flies, but it is not certain whether these came from the eggs or larvae themselves or from traces of media removed when collecting them.

Among the compounds specific to *D. melanogaster,*
*c*VA has previously been shown to reduce mating in *D. suzukii* [30], and (*Z,Z*)-7,11-heptadecadiene deterred mating in *D. simulans* [33]. Distortion of the relative amounts of (Z)-5-, (Z)-7-, (Z)-9-tricosenes and tricosane disrupted mating in *D. suzukii* [31]. Nevertheless, application of hexane washes of adults, eggs, or larvae of *D. melanogaster* to media dishes had no effect on subsequent oviposition by *D. suzukii* in this study, indicating that the CHCs, or indeed any other hexane-soluble compounds, were not responsible for the effects observed with adults and larvae of *D. melanogaster*. These results are consistent with the findings that the reduction in oviposition by *D. suzukii* is only observed in the presence of live larvae of *D. melanogaster* or after their recent removal. Furthermore, we ascertained that *D. melanogaster* larvae had to be alive, whereby the presence of dead *D. melanogaster* larvae alone did not deter *D. suzukii* from ovipositing.

Other potential candidates for the deterrent signal left by *D. melanogaster* are metabolites from microbes associated with *D. melanogaster* larvae and the larval frass. Volatiles from *D. melanogaster* frass induced aggregation of conspecifics [39]. Microbes which include yeast and bacteria are found on and inside insect bodies. They can be species-specific, but may vary depending on their food source [40,41,42]. *Drosophila melanogaster* and *D. suzukii* were attracted towards food sources treated with yeast isolated from *Drosophila* gut [43,44,45]. Oviposition avoidance by *D. suzukii* was recorded on apple juice agar inoculated with microbes isolated from *D. melanogaster* and *Drosophila biamirpes,* respectively [46]. Fruits contaminated with faecal matter from non-*D. suzukii Drosophila* species were avoided by *D. suzukii* females for oviposition [37]. In a laboratory choice assay, *D. suzukii* showed a preference towards *Hanseniaspora uvarum,* a yeast species commonly associated with *D. suzukii* [47], and an acetic acid bacteria strain isolated from *D. suzukii* [48]. These examples showed that *Drosophila*-associated microbes can play a role in *D. suzukii* oviposition and attraction to food sources. However, we showed that *D. suzukii* oviposition was not affected on media dishes containing *D. melanogaster* larvae that were frozen in situ to kill the larvae (Figure 11). Microbes and their metabolites associated with *D. melanogaster* larvae or in the frass would be expected to remain active on the media even after being frozen, suggesting these may not play a major role in influencing *D. suzukii* oviposition, but further investigations are needed to confirm this.

Other factors influencing oviposition site selection by *D. suzukii* include fruit firmness [35], volatile, and visual cues [49,50,51]. Cornmeal media was used as the egg laying substrate in this study. Similar observations were made when fruits were used as the egg laying substrate, where *D. suzukii* laid fewer eggs on blueberry fruits previously visited and defecated on by *D. melanogaster* [37]. Whilst here we exposed the cornmeal media to *D. melanogaster* for 48 h, Kienzle and Rohlfs [37] only exposed blueberry fruits to *D. melanogaster* for 1.5 h, during which time they observed that *D. melanogaster* visited and defecated on the fruits but did not lay eggs. Further studies using wild-caught *D. suzukii* and fruit instead of cornmeal media and conducted in horticultural settings such as in glasshouses or polytunnels are planned but these are beyond the scope of this initial study. Established laboratory cultures of *D. suzukii* and *D. melanogaster* were used in this study to make it comparable to our previous work [22] where our hypothesis originated from.

We recorded contrasting observations of the effect of the presence of conspecific larvae on *D. suzukii* oviposition: no effect on *D. suzukii* oviposition in Figure 9 and Figure 10, and reduction in *D. suzukii* oviposition in Figure 11. Notably, the reduction in *D. suzukii* oviposition in the presence of conspecific larvae in Figure 11 is not as extreme with *p*-value just below threshold at *p* < 0.05. Other studies showed that when *D. suzukii* encounters a food source previously visited by its conspecifics, the outcome on *D. suzukii* oviposition is indeed less consistent than the effects of prior visitation by other *Drosophila* species. Effects of conspecifics on *D. suzukii* egg laying varied from no impact [22], increased [51], or reduced egg laying [52]. These discrepancies are likely due to the difference in the egg laying media or fruit used, different laboratory strains of *D. suzukii*, and the length of time that female *D. suzukii* were allowed access to the test plates. Elsensohn et al. [52] exposed raspberry juice agar and raspberry puree agar to *D. suzukii* for 2 and 4 h, Tait et al. [51] checked the blueberry fruit at 2, 4, 24 and 48 h, whereas in both our current and previous study [22] we allowed *D. suzukii* 48 h access to the media. Elsensohn et al. [51] observed that female *D. suzukii′*s preference to lay eggs on unmarked media than on media marked by their conspecifics diminished after 4 h when the previously unmarked media becomes marked too. In contrast to *D. suzukii*, *D. melanogaster* females preferred to oviposit on sites where conspecific larvae were present [53,54]. The effect of conspecifics on *D. suzukii* oviposition is not in the scope of this work but it remains a valuable question to be investigated.

Lastly, whilst this study only used *D. melanogaster*, interspecific interactions with other *Drosophila* species are not always a disadvantage to *D. suzukii*. Certain interactions resulted in a neutral impact or no effect on *D. suzukii* oviposition. *Drosophila suzukii* exhibited no oviposition avoidance on substrates previously exposed to *Drosophila rufa* or *Drosophila auraria* [23]. In contrast, *D. suzukii* avoided ovipositing on substrates pre-inoculated with eggs from *D. melanogaster* or *Drosophila lutescens* [23]. This indicates that oviposition avoidance by *D. suzukii* from interaction with other *Drosophila* species is unlikely to be specific to *D. melanogaster* and will vary between species.

While we have not identified the actual signal from *D. melanogaster* that results in a reduction in oviposition by *D. suzukii*, we conclude that the presence of live *D. melanogaster* larvae on the egg laying media deters *D. suzukii* oviposition. *Drosophila melanogaster* CHCs were investigated as the egg laying deterrent signal but our results suggest that CHCs are unlikely to be the signal. Further exploration of this topic on insect-derived oviposition deterrent compounds or signals will be useful to optimize control methods against *D. suzukii* and to reduce reliance on chemical insecticides.

## Figures and Tables

**Figure 1 insects-13-00688-f001:**
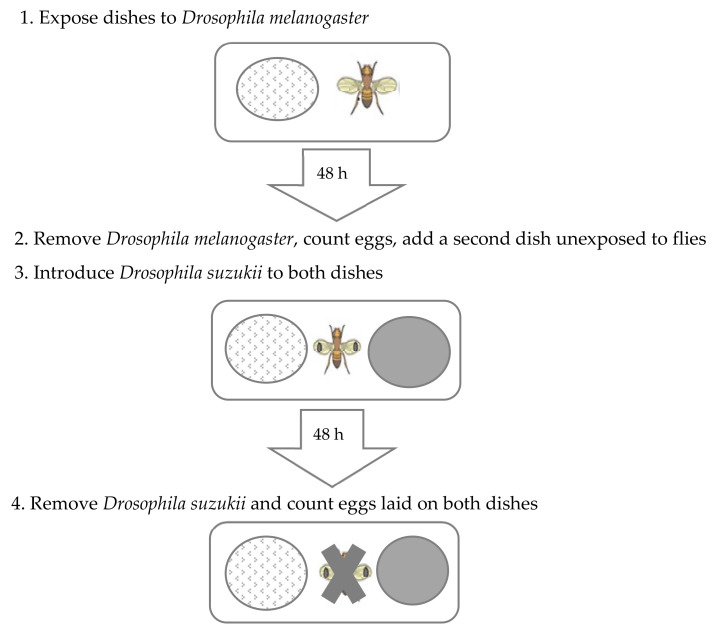
Schematic representation of the oviposition choice test experiment where *Drosophila suzukii* were given a choice to oviposit on cornmeal media dish that had been pre-exposed to *D. melanogaster* versus media that had never been exposed to *D. melanogaster* (blank) (adapted from Shaw et al. [22]).

**Figure 2 insects-13-00688-f002:**
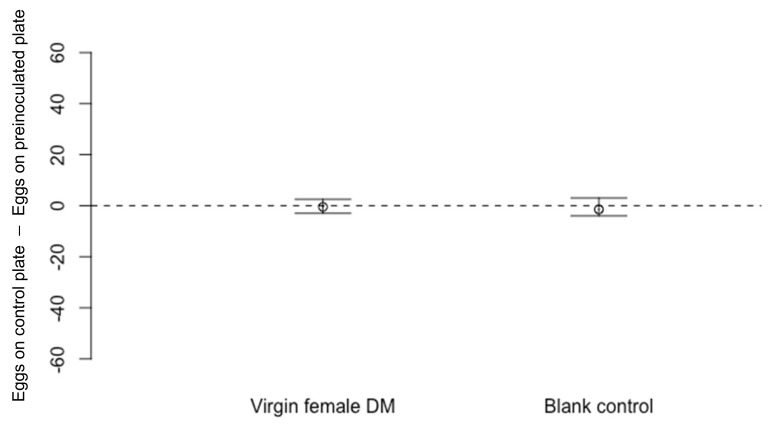
Differences in numbers of eggs laid by *Drosophila suzukii* (Median ± 95% confidence intervals) on cornmeal media dishes previously exposed for 48 h to virgin *D. melanogaster* (DM) females cmpared to a blank dish not previously exposed to flies (*n* = 17, *p* = 0.83). In the control experiment, *D. suzukii* laid a similar number of eggs on both blank dishes that were not exposed to *D. melanogaster* previously (*n* = 17, *p* = 0.67).

**Figure 3 insects-13-00688-f003:**
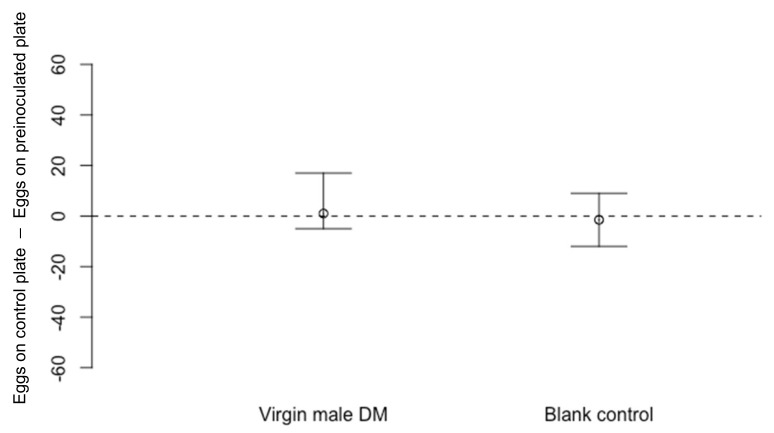
Differences in numbers of eggs laid by *Drosophila suzukii* (Median ± 95% confidence intervals) on cornmeal media dishes previously exposed for 48 h to virgin *D. melanogaster* (DM) males compared to a blank dish not previously exposed to flies (*n* = 10, *p* = 0.72). In the control experiment, *D. suzukii* laid a similar number of eggs on both blank dishes that were not exposed to *D. melanogaster* previously (*n* = 10, *p* = 0.77).

**Figure 4 insects-13-00688-f004:**
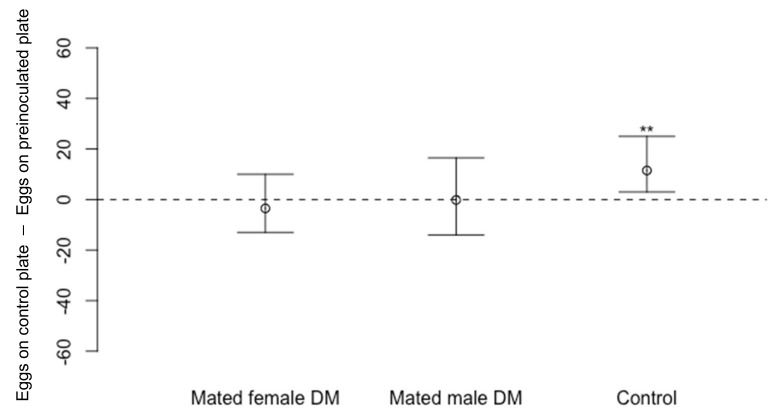
Differences in numbers of eggs laid by *Drosophila suzukii* (Median ± 95% confidence intervals) on cornmeal media dishes previously exposed to mated *D. melanogaster* (DM) females (*n* = 16, *p* = 0.50) or males (*n* = 16, *p* = 1.00) compared to a blank dish not previously exposed to *D. melanogaster*. In the control experiment, *D. suzukii* laid fewer eggs on cornmeal media dishes previously exposed to male and female *D. melanogaster* compared to blank dishes (*n* = 16, *p* < 0.01 ****).

**Figure 5 insects-13-00688-f005:**
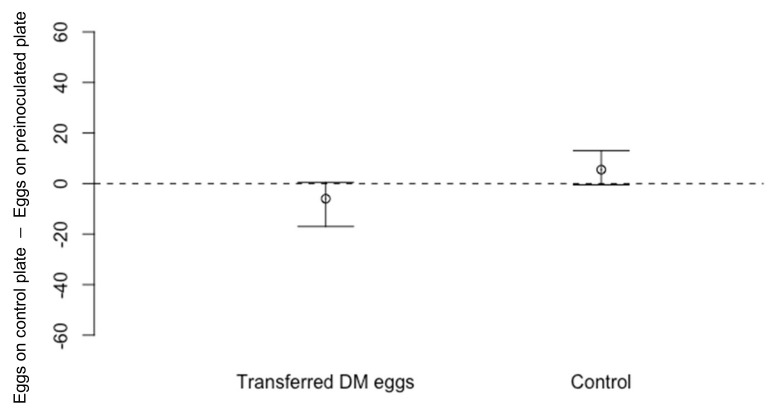
Differences in numbers of eggs laid by *Drosophila suzukii* (Median ± 95% confidence intervals) laid on cornmeal media dishes which contained transferred *D. melanogaster* (DM) eggs compared to eggs laid on a blank dish (*n* = 20, *p* = 0.07). In the control experiment, *D. suzukii* laid fewer eggs on dishes pre-exposed to *D. melanogaster* compared to unexposed dish, but this difference was not statistically significant (*n* = 20, *p* = 0.07).

**Figure 6 insects-13-00688-f006:**
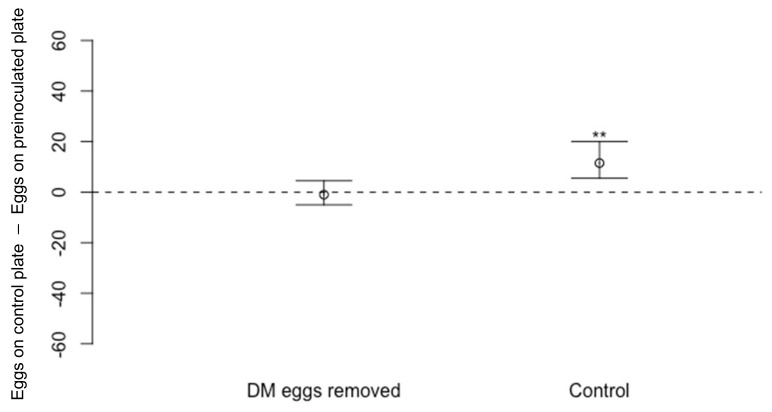
Differences in numbers of eggs laid by *Drosophila suzukii* (Median ± 95% confidence intervals) laid on cornmeal media dishes where *D. melanogaster* (DM) females were allowed to lay eggs in which the eggs were then removed compared to *D. suzukii* eggs laid on a blank dish (*n* = 20, *p* = 0.64). In the control experiment, fewer eggs were laid by *D. suzukii* on cornmeal media dishes pre-exposed to *D. melanogaster* compared to those on blank, unexposed dish (*n* = 20, *p* < 0.01 **).

**Figure 7 insects-13-00688-f007:**
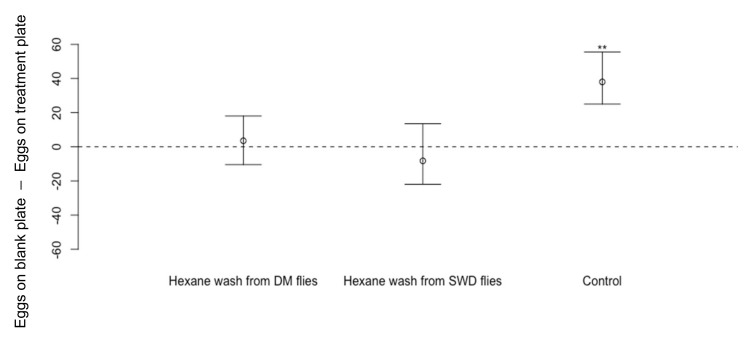
Differences in numbers of eggs laid by *Drosophila suzukii* (Median ± 95% confidence intervals) on cornmeal media dishes treated by hexane washes from whole bodies of *D. melanogaster* (DM) (*n* = 12, *p* = 0.61) or *D. suzukii* (SWD) (*n* = 12, *p* = 0.50) compared to eggs laid on blank cornmeal media dishes treated only with hexane. In the control experiment, *D. suzukii* laid fewer eggs on cornmeal media dishes pre-exposed to *D. melanogaster* for 48 h beforehand (*n* = 12, *p* < 0.01 **).

**Figure 8 insects-13-00688-f008:**
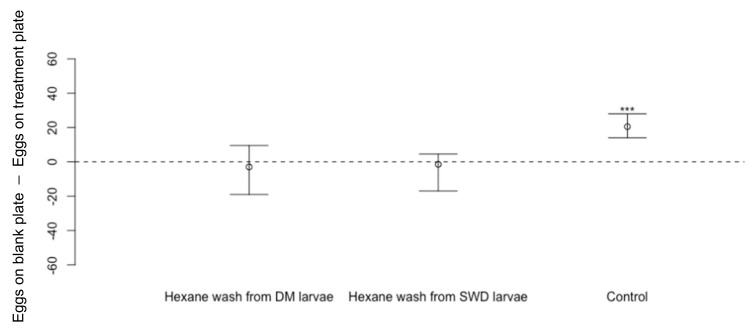
Differences in numbers of eggs laid by *Drosophila suzukii* (Median ± 95% confidence intervals) on cornmeal media dishes treated by hexane washes from larvae of *D. melanogaster* (DM) (*n* = 12, *p* = 0.62) or *D. suzukii* (*n* = 12, *p* = 0.54) compared to eggs laid on blank cornmeal media dishes treated only with hexane). In the control experiment, *D. suzukii* laid fewer eggs on cornmeal media dishes pre-exposed to *D. melanogaster* for 48 h beforehand (*n* = 12, *p* < 0.001 ***).

**Figure 9 insects-13-00688-f009:**
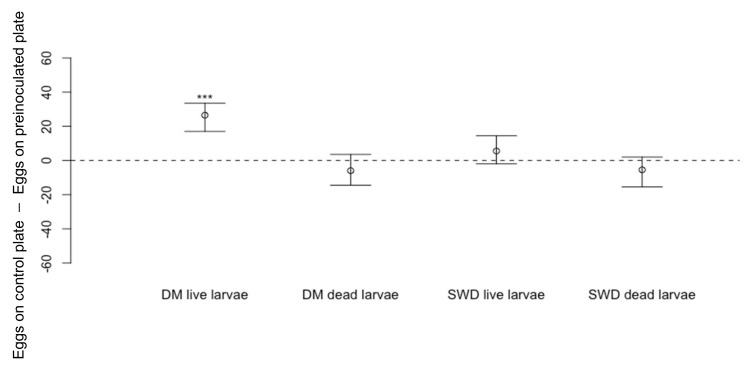
Differences in numbers of eggs laid by *Drosophila suzukii* (Median ± 95% confidence intervals) laid on cornmeal media dishes which contained live or dead larvae of *D. melanogaster* (DM: *n* = 18, *p* < 0.001 *** and *n* = 18, *p* = 0.28, respectively) or *D. suzukii* (SWD: *n* = 18, *p* = 0.13 and *n* = 18, *p* = 0.14, respectively) compared to numbers on blank cornmeal media dishes.

**Figure 10 insects-13-00688-f010:**
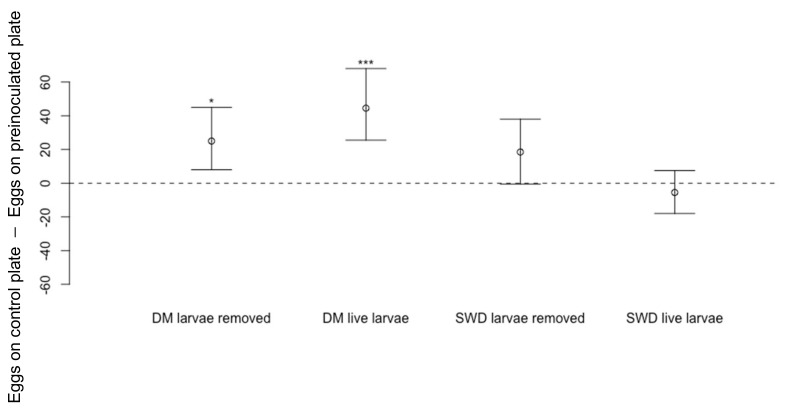
Differences in numbers of eggs laid by *Drosophila suzukii* (Median ± 95% confidence intervals) on cornmeal media dishes previously fed upon by *D. melanogaster* (DM) larvae for 48 h and then removed (*n* = 18, *p* < 0.05 *) or containing live larvae of *D. melanogaster* (*n* = 18, *p* < 0.001 ***) compared to numbers laid on a blank dish. There were no significant differences between the numbers of *D. suzukii* eggs recorded on cornmeal media dishes previously fed on by *D. suzukii* (SWD) larvae (*n* = 18, *p* = 0.06) or containing live *D. suzukii* larvae (*n* = 18, *p* = 0.45).

**Figure 11 insects-13-00688-f011:**
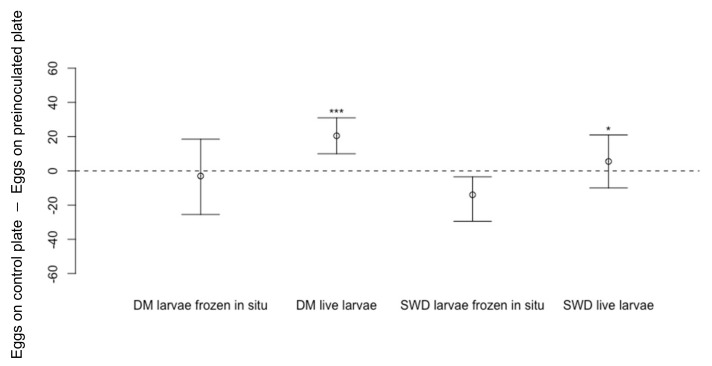
Differences in numbers of eggs laid by *Drosophila suzukii* (Median ± 95% confidence intervals) on cornmeal media dishes with larvae killed by freezing in situ of *D. melanogaster* (DM) (*n* = 18, p = 0.62) or *D. suzukii* (SWD) (*n* = 18, *p* = 0.41). Numbers of eggs laid by *D. suzukii* were lower on dishes containing live *D. melanogaster* (DM) larvae (*n* = 18, *p* < 0.001 ***) or live *D. suzukii* (SWD) larvae (*n* = 18, *p* < 0.05 *) compared to eggs laid on blank dishes.

**Figure 12 insects-13-00688-f012:**
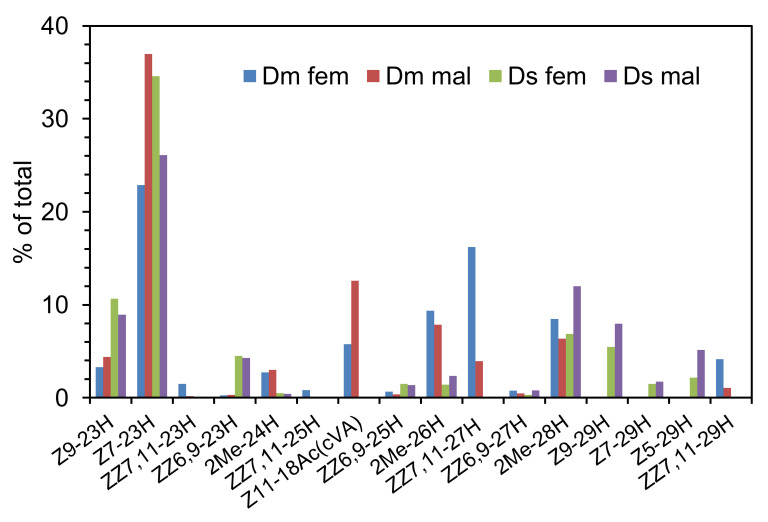
Relative amounts (% of total) of compounds identified in hexane extracts of adult females (Dmfem) and males (Dmmal) of *Drosophila melanogaster* and females (Dsfem) and males (Dsmal) of *D. suzukii.* See text for compound abbreviations.

## Data Availability

All the data included in this study can be found in this manuscript and its Appendix A.

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
