# Peer review of "Live Drosophila melanogaster Larvae Deter Oviposition by Drosophila suzukii"

_insects, 2022, doi:10.3390/insects13080688_

Round 1

Reviewer 1 Report

This revised manuscript entitled "Live Drosophila melanogaster Larvae deter Oviposition by Drosophila suzukii"  describes an interesting study about the relationship between presence of Drosophila melanogaster and oviposition in Drosophila suzukii. They first showed that the number of eggs laid by D. Suzukii is decreased when co-incubated with D. Melanogaster. To further determine which stage of D. Melanogaster deter oviposition of D. Suzukii. The authors designed a two-choice assay combined with adult, larval and eggs from D. Melanogaster. The authors found that fewer eggs were laid by D. Suzukii when cultured with D. Melanogaster larvae. Finally, the authors examined cuticular hydrocarbons (CHC) from adult D. Melanogaster to check which external component is important for oviposition deterrent, they found no clear relationship between CHC and oviposition in D. Suzukii.   

Overall, it is an interesting paper and worth to be acknowledged by readers. in addition, I'm pleased to see that comments/questions raised by reviewers in the first round have been properly addressed. I have no further questions prior to publication.

Author Response

Dear Reviewer 1, thank you for your constructive comments and for reviewing our manuscript. We were pleased that we have been able to fulfil the comments/questions raised by reviewers in the first round.

Reviewer 2 Report

Greetings and respect

In the introduction, D. suzukii and D. Melanogaster have been correctly described in the introduction. But it is not enough. Adding these details would improve the article. Authors should clearly explain why the research was done, why it was important, and how it fits with other studies. It should be clear and concise and it is not...

Author Response

Thank you for reviewing our manuscript. In response to your query that our introduction section did not explain clearly on : why the research was done, why it was important, and how it fits with other studies – we have made edits to the introduction section to better explain these parts. We have edited and highlighted the introduction section with track changes where those points were described.

We edited parts of the introduction section to improve the clarity on why the research was done and why it was important (L59, 60, 67, 71-72). The importance of Drosophila suzukii as a global invasive pest of soft- and stone-fruit production which causes significant economic loss is highlighted (L 55-57, 60-63). We then described the increasingly limited options of control for D. suzukii and risk of resistance developing from the repeated use of insecticides (L 65-72). The main aim of this research is to better understand what affect D. suzukii oviposition (L 71-74), how the knowledge can be used to optimise D. suzukii control methods.

We described how this study fits with other studies as detailed in the following. This study was done based on our earlier published findings in Shaw et al., (2018) where we recorded that D. suzukii laid fewer eggs on media that were previously visited by another Drosophila species, D. melanogaster (L 81-84). In nature, D. suzukii will encounter D. melanogaster at the end of the season where most fruits have been harvested as D. melanogaster feeds and oviposit on rotten fruit (L 78-83). The outcome of interaction between D. suzukii and other Drosophila species in nature is not yet widely understood (L 82-83). Another recent study by Kidera and Takahashi (2020) reported a similar finding to ours that D. suzukii laid fewer eggs on D. melanogaster pre-visited media (L 85) but it is not yet know what causes this. All this led us to form our hypothesis for this study (L 87-90) and how this work can contribute to improve D. suzukii control methods (L 90-93). Thank you for your time and we hope that this is sufficient to improve the clarity of this work.

This manuscript is a resubmission of an earlier submission. The following is a list of the peer review reports and author responses from that submission.

Round 1

Reviewer 1 Report

Title of the study: “Live Drosophila melanogaster Larvae deter Oviposition by Drosophila suzukii

This study investigates which life-stage of Drosophila melanogaster affects the oviposition of Drosophila suzukii. In a series of laboratory-based choice tests, the authors demonstrated that the presence of live D. melanogaster larvae on the egg-laying media had a deterrent effect on D. suzukii oviposition. The study is comprehensive and well-written. Even though the results and statistical analysis are truthfully executed in answering the research questions, fewer improvements will provide a better understanding of this section to the readers. The conclusion provided by this research will give the opportunity for further studies aiming to identify the signals responsible for the oviposition reduction of D. suzukii, therefore, interesting, and worth publishing. Please find below some minor suggestions to improve it.

 Lines 83-84. Please correct: “Drosophila suzukii is able to detect CHCs from other Drosophila species which affect its behavior. For example, …”.

Lines 100-102. Please revised the percentages of the ingredients of the standard cornmeal media in order to obtain 100 % totally.

Line 108. Please add the relative humidity condition (RH) if known.

Line 109. 2.2. Please name the paragraph: “General Procedure Oviposition Choice Tests”.

Line 264-266. Please correct the sentence, the word “treated” is repeated more times than needed.

Lines 287-289. Please rephrase. The sentence is confusing as in the Control experiment there should be no pre-exposed dishes.

Fig. 4. Same as above. Is the “Control” mentioned in the Fig.4 an average of the blank control dishes used in the “CONTROL” experiment run simultaneously or the average of the two “BLANK CONTROL” petri dishes each used in the mated female or male experiments? Please rephrase the Figure’s caption.

Please state clearly in the text the difference between the “blank control” dishes (=not previously exposed to flies but placed in the same ventilated Perspex boxes as the pre-exposed dishes) and the “control” (= blank dishes placed in a control experiment run simultaneously). Then please modify consequently these terms along in all manuscript and figures.

Reviewer 2 Report

The authors studied the effects of preconditioning by Drosophila melanogaster on oviposition of D. suzukii. I have some questions.

1) During laboratory culture, flies usually adapt to laboratory conditions. In D. suzukii, for example, newly established strains are rather difficult to rear with laboratory medium (probably because females lay fewer eggs and/or larval development is retarded), but their performance is gradually improved in later generations probably due to adaptation to laboratory conditions. I think it is not appropriate to use old strains to examine adaptive traits such as oviposition preference.

2) As the authors note, D. suzukii mainly oviposits in healthy fruits and rather exceptionally oviposits in rotten fruits in which D. melanogaster mainly oviposits. In addition, there are a number of Drosophila species and other insects that oviposit in rotten fruits. I think it is rather unreasonable to hypothesize that D. suzukii evolves a means to specifically detect the presence of D. melanogaster larvae using its cuticular hydrocarbons as a cue. For example, changes of resource (fruit) conditions due to larval activities would be more general cues for the presence of competitor larvae. I think that the modification of medium by larval activities should be tested before examining cuticular hydrocarbonds. In addition, not artificial medium but fruits should be used in the test.

3) Lines 157-160: According to this procedure, D. melanogaster eggs were left 96 h at most in experiments examining the effect of pre-exposure to mated D. melanogaster females or eggs. In 96 h, at least all D. melanogaster eggs hatch and then medium surface would become sloppy because of larval activities. Is it possible to count the number of eggs and egg envelops exactly? I think it is not easy to find all egg envelops.

4) Lines 169-173: Were these 15 females or males left in a box for 48 h for pre-exposure?

5) Figure 4: In my understanding, the experiment with “mated female DM” differs little from the “control” experiment (the difference is only the presence or absence of males in pre-exposure”). Why did the results differ?

The authors studied the effects of preconditioning by Drosophila melanogaster on oviposition of D. suzukii. I have some questions.

1) During laboratory culture, flies usually adapt to laboratory conditions. In D. suzukii, for example, newly established strains are rather difficult to rear with laboratory medium (probably because females lay fewer eggs and/or larval development is retarded), but their performance is gradually improved in later generations probably due to adaptation to laboratory conditions. I think it is not appropriate to use old strains to examine adaptive traits such as oviposition preference.

2) As the authors note, D. suzukii mainly oviposits in healthy fruits and rather exceptionally oviposits in rotten fruits in which D. melanogaster mainly oviposits. In addition, there are a number of Drosophila species and other insects that oviposit in rotten fruits. I think it is rather unreasonable to hypothesize that D. suzukii evolves a means to specifically detect the presence of D. melanogaster larvae using its cuticular hydrocarbons as a cue. For example, changes of resource (fruit) conditions due to larval activities would be more general cues for the presence of competitor larvae. I think that the modification of medium by larval activities should be tested before examining cuticular hydrocarbonds. In addition, not artificial medium but fruits should be used in the test.

3) Lines 157-160: According to this procedure, D. melanogaster eggs were left 96 h at most in experiments examining the effect of pre-exposure to mated D. melanogaster females or eggs. In 96 h, at least all D. melanogaster eggs hatch and then medium surface would become sloppy because of larval activities. Is it possible to count the number of eggs and egg envelops exactly? I think it is not easy to find all egg envelops.

4) Lines 169-173: Were these 15 females or males left in a box for 48 h for pre-exposure?

5) Figure 4: In my understanding, the experiment with “mated female DM” differs little from the “control” experiment (the difference is only the presence or absence of males in pre-exposure”). Why did the results differ?

Reviewer 3 Report

The paper written by Tungadi and colleagues describes an interesting study about the relationship between presence of Drosophila melanogaster and oviposition in Drosophila suzukii. They first showed that the number of eggs laid by D. Suzukii is decreased when co-incubated with D. Melanogaster. To further determine which stage of D. Melanogaster deter oviposition of D. Suzukii. The authors designed a two-choice assay combined with adult, larval and eggs from D. Melanogaster. The authors found that fewer eggs were laid by D. Suzukii when cultured with D. Melanogaster larvae. Finally, the authors examined cuticular hydrocarbons (CHC) from adult D. Melanogaster to check which external component is important for oviposition deterrent, they found no clear relationship between CHC and oviposition in D. Suzukii.   

Overall, this is an important study as D. Suzukii is an invasive pest and cause fruit collapse and economic loss in agriculture. The study was well-designed, and the experiments listed in this paper were carefully performed. However, I have the following concerns regarding to the data presentation and analysis that need to be clarified prior to publication.

Major Concerns:

1.     The paper claims to identify the life stage of D. Melanogaster that affect egg-laying in D. Suzukii, but the pupal stage was not studied in this paper.

2.     The y-axis of all the egg-lying data (Fig2-9,11-12) was quite confusing, I suggest to use a performance index (PI=(eggs blank plate-eggs treatment plate)/ eggs blank plate)  to replace it.

3.     In Figure 4, I don’t understand why there is a difference between the control and mated female, more discussion is needed.

4.     In figure 5 and 6, are the controls between fig5 and 6 from the same flies or two in parallel experiments? I suggest combining them together to make the paper neat.

5.     Figure10, if the larval stage is important, why CHC from the adult flies were tested here?

6.     Also, cVA was considered as a male specific pheromone, why is it also found in female flies in fig10?

7.     Fig9, is there any statistical difference between DM larvae and SWD larvae?

8.     “The control” listed in fig 4,5,6 11 and 12 looks quite inconsistent, please explain why.